# DERO: Diffusion-Model-Erasure Robust Watermarking

## ABSTRACT

The effective denoising demonstrated by the latent diffusion model poses a new threat to image watermarking, as attackers can erase the watermark by performing a forward diffusion, followed by backward denoising. While such denoising might introduce large distortion in the pixel domain, the image semantics remain similar. Unfortunately, most existing robust watermarking methods fail to tackle such an erasure attack since they are primarily designed for traditional channel distortions. To address such issue, this paper proposed DERO, a **d**iffusion-model-**e**rasure **ro**bust watermarking framework. Based on the frequency domain analysis of the diffusion model's denoising process, we designed a destruction and compensation noise layer (DCNL) to approximate the distortion effects caused by latent diffusion model erasure (LDE). In detail, DCNL consists of a multi-scale low-pass filtering and a white noise compensation process, where the high-frequency components of the image are first obliterated, and then full-frequency components are enriched with white noise. Such a process broadly simulates the LDE distortions. Besides, on the extraction side, we cascaded a pre-trained variational autoencoder before the decoder to extract the watermark in the latent domain, which closely adapts to the operation domain of the LDE process. Meanwhile, to improve the robustness of the decoder, we also design a latent feature augmentation (LFA) operation on the latent feature. Throughout the end-to-end training with the DCNL and LFA, DERO can successfully achieve robustness against LDE. Our experimental results demonstrate the effectiveness and the generalizability of the proposed framework. The LDE robustness is significantly improved from 75% with SOTA methods to an impressive 96% with DERO.

## CCS CONCEPTS

• **Security and privacy** → **Digital rights management**.

## KEYWORDS

Diffusion model, robust watermarking, watermark erase

**ACM Reference Format:**
Anonymous Author(s). 2024. DERO: Diffusion-Model-Erasure Robust Watermarking. In *Proceedings of ACM Multimedia (ACM MM)*. ACM, New York, NY, USA, 9 pages. https://doi.org/XXXXXXX.XXXXXXX

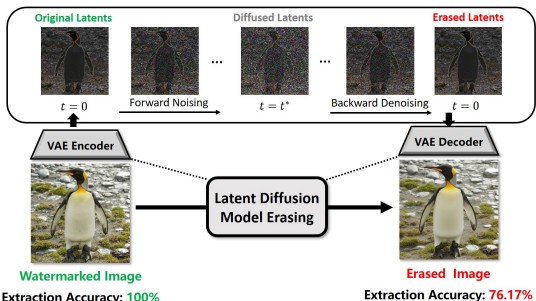

**Figure 1: The process and influence of LDE on watermarked images, evaluated with StegaStamp [26]. After LDE, the watermark cannot be extracted accurately.**

## 1 INTRODUCTION

Deep learning-based image watermarking mechanisms [1, 5, 14, 15, 18, 19, 26, 33] have garnered significant attention in recent research. These mechanisms typically rely on an architecture known as "Encoder-Noise Layer-Decoder". In this setup, the encoder's purpose is to embed the watermark into the host image, the noise layer introduces deliberate distortions into the image, and the decoder tries to extract the embedded watermark signal from the distorted images. Among these components, the critical factor for achieving robustness lies in the effective design of the noise layer.

Many methods have been proposed to address diverse attack scenarios. The majority of them focus on transmission distortions, for example, JPEG-Mask [33] and MBRS[10] tailored for the robustness of JPEG compression; Stegastamp [26] tailored for print-to-camera scenarios, and PIMoG [5] tailored for screen-to-camera robustness.

However, beyond transmission distortions, watermarked images may also be subjected to deliberate erasure. Previously, designing a high-quality watermark erase mechanism needed massive "host-watermarked" image pairs [7, 27]. However, the powerful denoising ability of the latent diffusion model provides feasibility for erasure mechanism design. One can easily leverage a black-box, openly accessible latent diffusion model API (*e.g.*, Stable Diffusion) to eliminate the embedded watermark signal by first adding noise and then conducting the denoising process on the noised latent. Due to the powerful denoising capability of the latent diffusion model, the watermark signal can be efficiently removed from the watermarked image, while maintaining its semantic integrity and high visual quality, as shown in Fig. 1.

Our experimental results show the vulnerability of existing watermarking schemes to diffusion model erasing, including the state-of-the-art (SOTA) deep learning schemes and the classical watermarking schemes, as well as a well-known commercial watermarking system, Digimarc, as shown in Table 1. We randomly sampled 50 images from COCO datasets [13] and reshaped them to size 512×512 and embedded the watermark with length 256 bits into these images. Then we perform no distortion, Gaussian noise and latent diffusion model erasure (LDE) distortion (with Stable Diffusion v1.5 and denoising strength 0.15) on the watermarked images and test

**Table 1: The extraction accuracy with different watermarking mechanisms.**

| Method | HiDDeN[33] | MBRS[10] | DADW[14] | FIN[6] | CIN[15] | StegaStamp[26] | PIMoG[5] | LGDR[16] | DWTDFT[11] | Digimarc |
|---|---|---|---|---|---|---|---|---|---|---|
| Identity | 99.5% | 100% | 100% | 100% | 100% | 100% | 100% | 100% | 100% | ✓ |
| GaussianNoise | 85.34% | 99.37% | 85.60% | 97.26% | 97.28% | 99.34% | 99.58% | 100% | 93.05% | ✓ |
| LDE | 58.10% | 71.29% | 50.91% | 66.00% | 53.96% | 75.04% | 52.70% | 72.00% | 56.67% | ✗ |

the extraction performance. It can be seen that all the algorithms experienced a notable reduction in watermark extraction accuracy after LDE. Such a high-quality, low-barrier, easy-to-deploy attack leads to the goal of this paper, enhancing the robustness against LDE.

A trivial way to ensure the LDE robustness is by integrating the LDE process directly as a noise layer into the framework's training. However, due to the intricate nature of the diffusion model and the numerous sampling iterations involved, this approach imposes a substantial memory overhead for gradient computation. Consequently, it leads to time-consuming and inefficient training processes.

To address such limitations, we propose an alternative approach by introducing an LDE distortion approximation scheme as a noise layer during training. Building upon insights from [29] and our own observations, we note that in the generation process of the latent diffusion model, low-frequency components of an image are initially recovered, followed by gradual refinement of high-frequency components. Leveraging this observation, we designed the destruction and compensation noise layer (DCNL) to emulate the LDE distortions. Specifically, DCNL incorporates multi-scale low-pass filtering and a white noise compensation process, wherein the high-frequency components of the image are initially removed, followed by the addition of white noise to the full-frequency components. Consequently, the distortion introduced by LDE can be effectively approximated.

However, relying solely on DCNL may not suffice to guarantee robustness against LDE. Therefore, we introduce a complementary technique called latent feature augmentation (LFA) on the extraction side. Specifically, we utilize a pre-trained variational autoencoder (VAE) to encode the distorted images and apply LFA on the VAE-encoded features. LFA combines VAE-encoded features with Gaussian noise in a randomly weighted manner, which enables the decoder to be trained to extract watermarks from severely distorted latent representations, thereby enhancing the overall robustness of the system against LDE.

Based on DCNL and LFA, we proposed DERO, a **d**iffusion-model-**e**rasure **ro**bust watermarking framework. Additionally, by configuring different noise layers, the framework demonstrates excellent adaptability to various types of distortions.

The main contributions of this paper are summarized as follows:

1) We post a potential threat of latent diffusion model-based erasure (LDE) on watermarking systems and underscore the vulnerability of existing watermarking systems to LDE distortions.
2) Based on our analysis of the LDE process, which involves the initial recovery of low-frequency components followed by the gradual refinement of high-frequency components,

we have targetedly proposed a destruction and compensation noise layer (DCNL) to simulate the LDE distortion. By combining the proposed DCNL with a latent feature augmentation operation, we design DERO, a diffusion-model-erasure robust watermarking framework, which can be end-to-end trained to ensure robustness against LDE distortion.
3) Extensive experiments indicate the superior performance of the proposed framework on robustness for latent diffusion model erasure. Compared with the state-of-the-art DNN-based watermarking schemes, we observe a substantial improvement in extraction accuracy (from 80.89% with StegaStamp[26] to 98.49%).

## 2 RELATED WORKS

### 2.1 Deep learning-based watermarking

The current deep learning-based watermarking algorithm adopts an "Encoder-Noise Layer-Decoder" architecture, initially proposed by Zhu *et al.* [33]. By introducing various distortions within the noise layer, targeted robustness can be achieved. An essential requirement for the noise layer is its differentiability, and significant research efforts are directed toward designing differentiable operations that can effectively replicate non-differentiable distortions. For example, to ensure robustness against JPEG compression, Zhu *et al.* [33] introduced a differentiable operation called JPEG-Mask. Building upon this work, Jia *et al.* [10] developed a mini-batch-based noise layer to enhance JPEG robustness. In addition to addressing image editing distortions, some noise layers are tailored to physical distortions. To ensure print-shooting robustness, Tancik *et al.*[26] proposed a noise layer comprising several differentiable operations to replicate print-shooting distortions. Similarly, Fang *et al.*[5] introduced PIMoG, a noise layer designed for screen-shooting robustness. Furthermore, rather than utilizing specific operations, Luo *et al.*[14] suggested employing an adversarial network as the noise layer to generate the worst-case distorted image and iteratively train for improving robustness. As for the unknown distortions, a combined noise layer that contains multiple distortions is often employed[15]. However, the design of existing noise layers has predominantly concentrated on mitigating transmission distortions. In the case of LDE, none of these methods can be successfully applied.

### 2.2 Diffusion models

Diffusion models as well as the latent diffusion models [9, 20, 22, 24] have demonstrated remarkable performance in generative tasks, fitting applications across various computer vision domains, including inpainting, super-resolution, and even text-to-image generation [2, 4, 23]. Thanks to the exceptional sampling quality, diffusion models have emerged as the leading architecture for generation tasks

nowadays. In general, diffusion models encompass two primary processes: (1) the forward diffusion process, which introduces Gaussian noise on the image to perturb the input into random noise, and (2) the reverse process, which progressively transforms the random noise back into high-quality images through a step-by-step denoising procedure. The latent diffusion model typically uses variational autoencoder VAE to encode the image into latent features, and the processes of forward-diffusion and backward-denoising occur in the latent domain. The effectiveness of high-quality denoising in the reverse process has led to a series of intriguing developments in adversarial purifications[21]. Moreover, some advancements in the certifiable robustness of adversarial attacks are built upon the analysis of diffusion models [30]. It should be noted that the purification process is similar to watermark erasure, where the diffusion model is adept at erasing unexpected perturbations effectively. Such a capability poses a potential threat to the watermarking systems.

### 2.3 Diffusion model-based watermarking

Recently, many diffusion model-based watermarking schemes have been proposed [17, 25, 28, 31]. These methods typically follow a common procedure: given an image and its watermark, they first extract the diffused latent features of the image and then embed the watermark either within the latent features or during the denoising process. However, these approaches often fail to meet the visual quality requirements in the pixel domain for image watermarking. This limitation arises from the fact that latent domain embedding tends to preserve only semantic consistency, potentially leading to significant visual distortion in the pixel domain.

## 3 BACKGROUND

### 3.1 Denoising Diffusion Models

The denoising diffusion model (DDM) reverses a progressive noise process. Given the image sampled from the real distribution $x \in X$, it is noised with $T$ steps by Gaussian noise $\epsilon \sim N(0,1)$ with different schedules, shown as Eq. (1)

$$x_t = \sqrt{\alpha_t}x + \sqrt{1 - \alpha_t}\epsilon \qquad (1)$$

where $\alpha_t$ is a set of monotonic increasing scheduled timesteps $\{\alpha_t\}_{t=0}^{T}$, $\alpha_T = 0, \alpha_0 = 1$. The DDM $\epsilon_\theta$ is trained with the loss function $MSE(\epsilon_\theta(x_t, t), \epsilon)$ to predict the added noise on each timesteps. At the backward denoising process, given a noise vector $x_T$, the noise is progressively diminished through sequential predictions made by $\epsilon_\theta$ over $T$ steps. The most common sampling scheme is DDIM [24] where intermediate steps are calculated as:

$$x_{t-1} = \sqrt{\frac{\alpha_{t-1}}{\alpha_t}}x_t + \left(\sqrt{\frac{1}{\alpha_{t-1}} - 1} - \sqrt{\frac{1}{\alpha_t} - 1}\right) \cdot \epsilon_\theta(x_t, t) \quad (2)$$

In the current text-guided diffusion models, DDM $\epsilon_\theta$ also contains another input which is text condition $C$ to generate the image corresponding to the given conditioning prompt, i.e., $\epsilon_\theta(x_t, t, C)$. For the latent diffusion model [22], the image is first encoded by a pre-trained VAE and all the diffusion and denoising operations are performed on the latent features $z_t = \mathcal{E}_{VAE}(x_t)$.

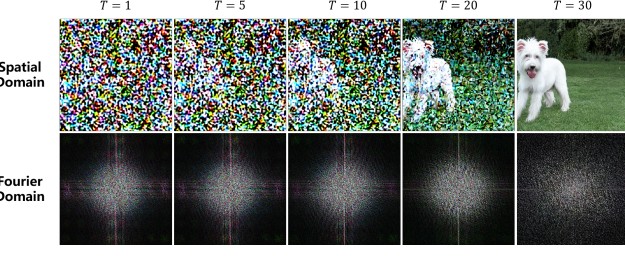

**Figure 2: The generated images and the corresponding frequency components with different timesteps.**

### 3.2 Latent Diffusion Model Erasure

Here we described the latent diffusion model erasure process. For the watermarked images $x^w$, the attacker can first encode with the VAE encoder of the LDM to get the latent features $z^w = \mathcal{E}_{VAE}(x^w)$. Then Gaussian noise of zero mean and certain variance $\tilde{\epsilon}$ is added to $z^w$ to get the noisy latent $z_N^w = z^w + \tilde{\epsilon}$. Subsequentially, $z_N^w$ is treated as the starting point and is denoised by LDM $\epsilon_\theta$ with $\tilde{T}$ steps. The denoised latent $z^{\tilde{w}}$ is then decoded by VAE decoder to get the erased image $x^e = \mathcal{D}_{VAE}(z^{\tilde{w}})$. Such an erasure process is easy to automatically deploy by choosing "img2img" functions in the "WebUI" of Stable Diffusion [1], details can be found in the supplementary materials.

## 4 PROPOSED FRAMEWORK

### 4.1 Motivation and Analysis

The key to ensuring LDE robustness is designing a noise layer capable of accurately representing the LDE distortion. To achieve this objective, we begin by analyzing the denoising process of the latent diffusion model from a frequency domain perspective. We record the generation process of the latent diffusion model with different timesteps in both spatial domain and Fourier domain, as shown in Fig. 2. It is clear that with the time steps, more details of the generated image are refined in the spatial domain. From the Fourier perspective, low-frequency components are first generated, then high-frequency components are gradually completed. Meanwhile, during the refinement of the high-frequency component, changes are also observed in the low-frequency part. Based on the observation, we design the destruction and compensation noise layer, which characterizes the LDE distortion through a model involving low-pass filtering followed by full-band component compensation. Nevertheless, such an approximation is not sufficient to model all the distortions generated by the LDE. To address this limitation, we additionally perform latent feature-level augmentation to introduce further distortions in training the decoder.

### 4.2 Architecture

The framework of the proposed DERO is shown in Fig. 3. In the training stage, the host image $I_h \in \mathbb{R}^{C \times H \times W}$ and the watermark $w \in \{0, 1\}$ are first fed into the encoder **E** which outputs the watermarked image $I_w \in \mathbb{R}^{C \times H \times W}$. Then $I_w$ is distorted by DCNL to get the distorted image $I_d \in \mathbb{R}^{C \times H \times W}$. Subsequentially, $I_d$ is first encoded by a pre-trained VAE encoder $\mathcal{E}_{VAE}$ to get the latent feature $L_d \in \mathbb{R}^{C \times H/8 \times W/8}$, which is further augmented by LFA to

---
[1]https://github.com/AUTOMATIC1111/stable-diffusion-webui

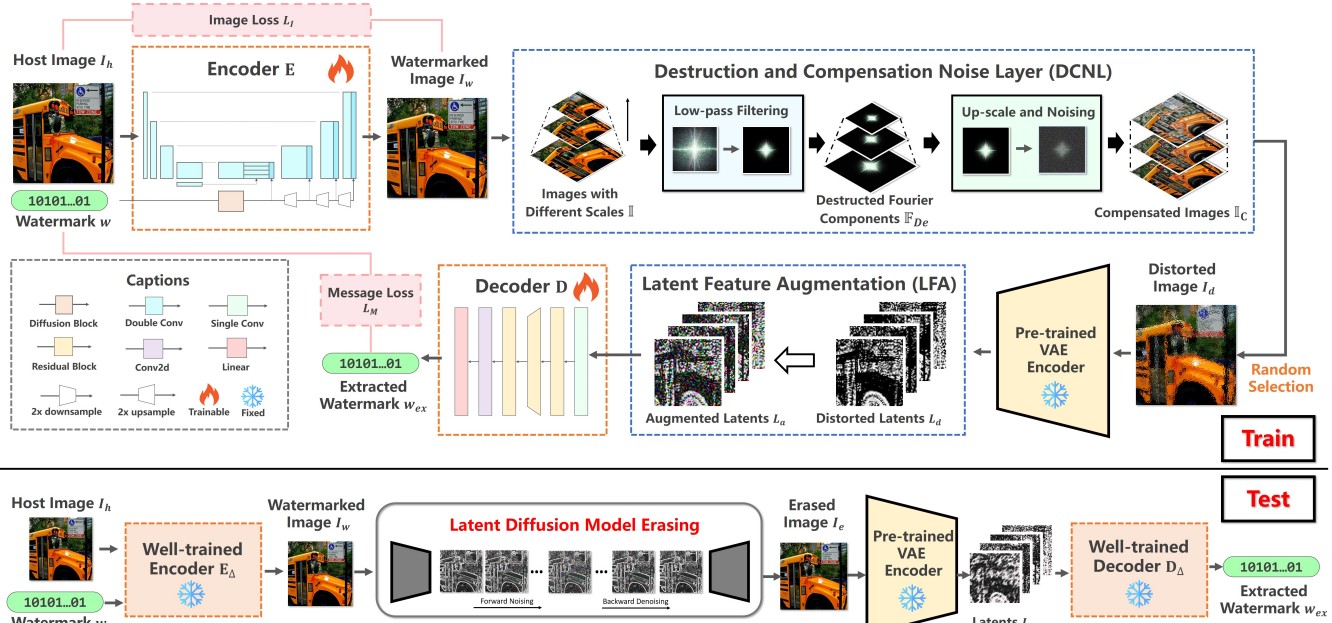

**Figure 3: The proposed framework. It consists of four main parts: the encoder E, the destruction and compensation noise layer, the latent feature augmentation and the decoder D. The whole framework is trained end-to-end with two loss functions which constrains the visual similarity of the host image and the watermarked image and the extraction accuracy of the embedded watermark.**

generate the augmented latents $L_a \in \mathbb{R}^{C \times H/8 \times W/8}$. The decoder **D** extracts the watermark $w_{ex}$ from $L_a$. While in the testing stage, the well-trained encoder $\mathbf{E}_\Delta$ and decoder $\mathbf{D}_\Delta$ are fixed. When receiving the erased/distorted image $I_e$, we first use $\mathcal{E}_{VAE}$ (same as the training stage) to get the latent features $L_{ex}$ and then feed $L_{ex}$ into $\mathbf{D}_\Delta$ for the final extraction.

*4.2.1 Encoder.* The encoder **E** aims to embed the watermark into the host image. In this paper, we utilize a UNet-based architecture as the backbone of **E**. Specifically, four "double conv" blocks (the concatenation of two "Conv-BN-ReLU" blocks) are first applied to downsample $I_h$ to the feature with size of $H/32 \times W/32$. At the same time, one linear layer is applied to the watermark message to generate the watermark feature with the same size of $H/32 \times W/32$. Then four "up-conv" blocks (the concatenation of two "UpSampling-Conv-BN-ReLU" blocks) are applied to each hidden feature of **E**, which is further concatenated by the upsampled watermark features. The UNet-based architecture effectively facilitates the fusion of the watermark feature with the host image features across different scales, thereby enhancing the quality of the watermarked images.

*4.2.2 Destruction and Compensation Noise Layer.* The purpose of DCNL is to introduce a similar distortion as LDE to the watermarked images. Drawing from our observations of LDM generation, we devised a mechanism for destroying high-frequency components and compensating for full-frequency components. In detail, after obtaining the watermarked image $I_w$, we first down-sample it to generate the images with different scales $\mathbb{I}$, as shown in Eq. (3).

$$I_w^s = \mathcal{D}own(I_w, s) \tag{3}$$

where $\mathcal{D}own$ indicates the down-sampling operation and $s$ is the scale of the down-sampling. $\mathbb{I} = \{I_w^0, I_w^1, I_w^2, ...\}$. For each $I_w^s$, we

apply discrete Fourier transformation (DFT) with Eq. (4).

$$F_w^s(U, V) = \sum_{u=1}^{H} \sum_{v=1}^{W} I_w^s(u, v) e^{-j2\pi \left(\frac{U}{H}u + \frac{V}{W}v\right)} \tag{4}$$

where $I_w^s(u, v)$ is the pixel value at $(u, v)$ in spatial domain and $F_w^s(U, V)$ is the complex value of $(U, V)$ in frequency domain. Then a low-pass filtering process is performed on each Fourier coefficients $F_w^s$ with Eq. (5).

$$F_{De}^s(U, V) = \mathcal{H}(U, V, s) \cdot F_w^s(U, V) \tag{5}$$

where

$$\mathcal{H}(U, V, s) = e^{-\frac{D^2(U,V)}{2D_s^2}} \tag{6}$$

$D(u, v) = u^2 + v^2$, $D_s$ indicates the cutoff frequency of scale $s$. So in total, we can get multiple low-pass filtered frequency components with different scales $\mathbb{F}_{De} = \{F_{De}^0, F_{De}^1, F_{De}^2...\}$. After that, an up-sampling operation with scale $s$ is first applied on $\mathbb{F}_{De}$, then white Gaussian noises $\xi$ with variance $\sigma$ are added into each upsampled frequency component to get the compensated Fourier coefficients $\mathbb{F}_C = \{F_C^0, F_C^1, F_C^2...\}$, where

$$F_C^s = \mathcal{U}p(F_{De}^s, s) + \xi \tag{7}$$

Given that the spectral density of white noise remains constant, implying its energy presence across the full band, we can effectively simulate the full band changing by adding white noise in the frequency domain. Then, by applying the inverse DFT on $\mathbb{F}_C$, we can get the compensated images $\mathbb{I}_C$.

*4.2.3 Latent Feature Augmentation.* Since the diffusion process in LDE is performed on the latent feature of the images, we proposed to extract the watermark in the latent domain to adapt to such distortion. Therefore, after obtaining $\mathbb{I}_C$, we randomly select one of them as the distorted image $I_d$, $I_d \in \mathbb{I}_C$. Then $I_d$ is encoded by a

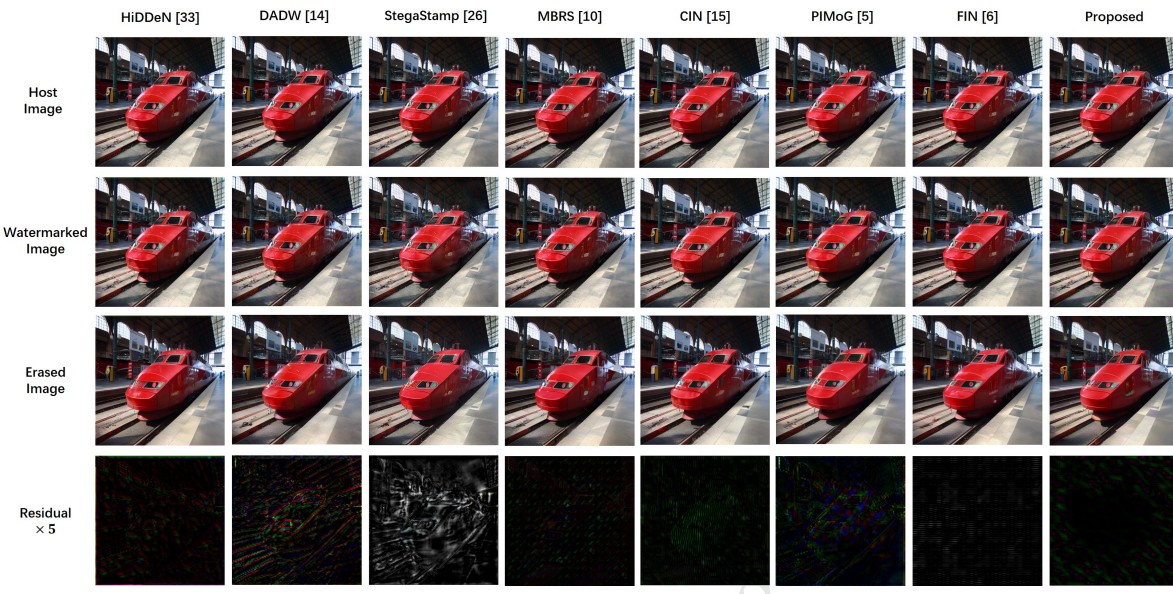

**Figure 4: The visual results of the watermarked image with different methods.**

pre-trained VAE encoder $\mathcal{E}_{VAE}$ to get the latent representation $L_d$. It is important to note that DCNL only serves as a rough simulation of the distortion introduced by LDE, which may not be sufficient to ensure robustness against LDE. To further enhance robustness, we apply augmentations on $L_d$. Specifically, $L_d$ is augmented through a weighted noising operation, as depicted in Eq. (8).

$$L_a = \alpha L_d + \beta \eta \tag{8}$$

where $\eta \sim N(0, 1)$ and is with the same size of $L_d$. $\alpha$ and $\beta$ indicates the weights. By performing LFA, the latent feature is further distorted, thereby presenting more stringent samples to enhance the extraction capability of the decoder. In this paper, $\alpha$ sampled uniformly from $[0.3, 0.5]$ and $\beta$ sampled uniformly from $[0.5, 0.8]$.

*4.2.4 Decoder.* The decoder **D** is designed to extract the watermark $w_{ex}$ from the augmented latent features $L_a$, which consists of one "single-conv" ("Conv-BN-ReLU") block, three "Res-Block" [8], one "Conv" block and one linear block. The downsampling operation is carried out in the "Res-Block".

### 4.3 Loss Functions

Two main loss functions are applied in training the whole framework, one is the image loss $\mathcal{L}_I$ which constrains the visual similarity of the watermarked image and the host image:

$$\mathcal{L}_I = \|I_h - I_w\|^2 = \|I_h - \mathbf{E}(I_h, w)\|^2 \tag{9}$$

Another loss function is the message $\mathcal{L}_M$ which ensure the extraction accuracy of the watermark:

$$\mathcal{L}_M = \|w - w_{ex}\|^2 = \|w - \mathbf{D}(L_a)\|^2 \tag{10}$$

Since all the procedures in DCNL and LFA are differentiable, the whole network can be trained end-to-end with $\mathcal{L}_I$ and $\mathcal{L}_M$ in the following manner:

$$\mathcal{L}_{total} = \lambda_I \mathcal{L}_I + \lambda_M \mathcal{L}_M \tag{11}$$

where $\lambda_I$ and $\lambda_M$ are the weight of the loss functions.

## 5 EXPERIMENTAL RESULTS

### 5.1 Implementation Details

*5.1.1 Dataset and Settings.* In this paper, MS COCO [13] is utilized as the training dataset for all stages. In the evaluation stage, 50 random images in the testing dataset were utilized. All images used, both in training and testing, have dimensions of 512×512×3. The length of the watermark message is set to 256 bits. The pre-trained VAE encoder $\mathcal{E}_{VAE}$ employed in this paper is the default VAE provided with Stable Diffusion v1.5. The entire framework is implemented using PyTorch [3] and executed on a Tesla V100 GPU. Adam [12] is utilized for parameter optimization.

*5.1.2 Baseline and Benchmark.* For the latent diffusion model erasure process, we utilize the "img2img" function in open-sourced Stable Diffusion web UI (SD-webUI). The specific procedure entails embedding the watermark message into the host image, followed by applying SD-webUI to erase the watermark. Upon erasure, we extract the watermark message from the erased image and record the extraction accuracy. The version of Stable Diffusion we use is v1.5[2]. The default image size is 512×512 with 30 sampling steps and a guidance scale of 7. All the comparison experiments are performed with these settings. To demonstrate the effectiveness of the proposed method, we compare its performance with seven state-of-the-art (SOTA) methods: HiDDeN [33], DADW [14], StegaStamp [26], MBRS [10], CIN [15], PIMoG[5] and FIN [6]. For HiDDeN, MBRS, FIN, and CIN, they were trained with the combined noise of JPEG compression, median filtering and Gaussian noise.

*5.1.3 Evaluation Metrics.* We evaluated the performance in two main aspects:

**Visual Quality of Watermarked Images**: Measured by the peak signal-to-noise ratio (PSNR), structural similarity index measure (SSIM) (where a larger value represents better visual quality), and

---

[2]https://huggingface.co/runwayml/stable-diffusion-v1-5

the LPIPS [32], which is a learned perceptual similarity metric (a lower value indicates better visual quality).

**Robustness Against LDE Distortion**: Assessed by the extraction accuracy of the watermark message after LDE distortion. Higher accuracy indicates stronger robustness.

## 5.2 Visual Quality

We provide one example to subjectively evaluate the visual quality of the watermarked image generated with different methods, as shown in Fig. 4. Additionally, we provide objective evaluations of visual quality through metrics (PSNR, SSIM, and LPIPS), as detailed in Table 2.

**Table 2: Visual quality of different methods.**

| Method | HiDDeN[33] | DADW[14] | StegaStamp[26] | MBRS[10] |
|---|---|---|---|---|
| **PSNR(dB)** | 36.60 | 33.07 | 28.55 | 39.60 |
| **SSIM** | 0.9647 | 0.9244 | 0.9329 | 0.9839 |
| **LPIPS** | 0.1358 | 0.2060 | 0.1525 | 0.0961 |
| **Method** | CIN [15] | PIMoG[5] | FIN [6] | DERO |
| **PSNR(dB)** | 40.21 | 38.32 | 39.19 | **40.53** |
| **SSIM** | 0.9845 | 0.9789 | 0.9820 | **0.9848** |
| **LPIPS** | 0.0833 | 0.0829 | 0.1207 | **0.0815** |

In Fig. 4, each column represents a different method. The first row, second row, third row, and bottom row correspond to the host images, watermarked images, erased images, and residuals, respectively. From the objective assessment results in Table 2, we can see that the proposed method maintains a high level of visual quality, where the PSNR value and SSIM value are higher than 40 dB and 0.98, respectively. Moreover, compared to the state-of-the-art (SOTA) methods, the proposed scheme achieves the best visual quality. All subsequent experiments assessing LDE robustness are conducted based on these visual quality results.

## 5.3 Comparison of the LDE Robustness

To comprehensively assess the robustness of LDE distortion, we utilized SD-webUI with varying denoising strengths (0.1/0.15/0.2) and different sampling methods (Eular, LMS, and DPM adaptive) for the diffusion erasure process. The robustness test results are presented in Table 3.

As shown in Table 3, it is evident that for all the compared methods, the latent diffusion model can successfully erase the watermark signal, even at a denoising strength of 0.1. Among them, the highest extraction rate is achieved by StegaStamp, reaching 80.89%. However, the proposed method consistently achieves high extraction accuracy, surpassing 97% at a denoising strength of 0.1. This significant performance gap underscores the superiority of the proposed framework. In addition, with regard to denoising strength, it is observed that as the strength increases, the watermark extraction accuracy decreases. This is because larger denoising strengths will result in stronger noise added to the watermarked latent, and consequently, a more pronounced erasure of the watermarked signal. Different sampling methods yield varied removal results, leading to distinct effects on the watermarked images. Notably, the proposed method demonstrates applicability across all tested sampling methods. Irrespective of denoising strengths and sampling methods, the

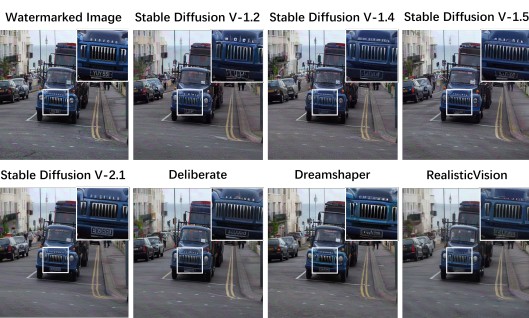

Watermarked Image  Stable Diffusion V-1.2  Stable Diffusion V-1.4  Stable Diffusion V-1.5

Stable Diffusion V-2.1  Deliberate  Dreamshaper  RealisticVision

**Figure 5: The visual results with different diffusion model versions.**

proposed method consistently exhibits the highest performance, with an extraction accuracy exceeding 94%, at least 17% higher than the compared methods. This outcome strongly underscores the effectiveness of the proposed noise layer and the robustness of the proposed method against LDE distortion.

## 5.4 Generalization of Robustness

*5.4.1 Different LDM Versions.* The advancement of diffusion model technology has led to the development of numerous fine-tuned models. Therefore, it is crucial to ensure LDE robustness across different versions. In this section, we primarily assess the generalizability of LDE robustness concerning model versions. Specifically, we gather seven different and commonly used versions (Stable Diffusion v1.2, v1.4, v1.5, v2.1, Deliberate (Deli), Dreamshaper (DS), and RealisticVision (RV)) for testing purposes. Subsequently, we conduct the LDE process with each of these versions. The visual results of the erased images (with sampling method "Eular" and strength 0.15) are presented in Fig. 5, and the corresponding PSNR and extraction accuracy are summarized in Table 4.

It is clear from Fig. 5 that the diffusion model with different versions will result in different erased images, primarily differing in details. Notably, when utilizing the "Dreamshaper" model, the color contrast becomes sharper, as observed in the zoom-in patches of the images. However, from the perspective of overall image quality, all models effectively maintain visual consistency with the watermarked image. This conclusion is further supported by Table 4, where the PSNR values of erased images with different versions of LDM are found to be at similar levels. Regarding LDE robustness, the proposed scheme consistently achieves a high extraction accuracy of at least 96%, demonstrating its effectiveness and resilience in the face of LDE distortions across different versions of the diffusion model. This underscores the robustness and reliability of the proposed approach.

*5.4.2 Different Sampling Settings.* In the diffusion process, the denoising result is affected by two main parameters: the sampling steps and the denoising strength. Different sampling steps will lead to different schedules of timestep, which will further influence the sampling weight $\alpha_t$ in each step. Different denoising strength gives different starting points to the latent sampling. Higher denoising strengths result in deeper denoising. In this section, we evaluate LDE robustness under different conditions by varying sampling steps and denoising strengths. We fixed the denoising strength at 0.15 and varied the sampling steps from 20 to 50 to conduct the test. Additionally, we fixed the sampling steps at 30 and tested with

**Table 3: Extraction accuracy of different methods after latent diffusion erasure distortions.**

| Sampling | Strengths | HiDDeN[33] | DADW[14] | Stegastamp[26] | MBRS[10] | CIN[15] | PIMoG[5] | FIN[6] | DERO |
|----------|-----------|------------|----------|----------------|----------|---------|----------|--------|------|
| Eular | 0.1 | 59.41% | 51.02% | 80.89% | 72.98% | 52.73% | 55.02% | 71.37% | **98.49%** |
| | 0.15 | 58.10% | 50.91% | 75.04% | 71.29% | 52.70% | 53.90% | 66.00% | **98.25%** |
| | 0.2 | 55.98% | 50.76% | 67.17% | 64.93% | 52.72% | 52.86% | 59.75% | **96.92%** |
| LMS | 0.1 | 59.36% | 50.88% | 80.44% | 73.92% | 52.75% | 55.07% | 78.47% | **98.78%** |
| | 0.15 | 58.13% | 50.94% | 78.5% | 71.94% | 52.71% | 54.90% | 74.53% | **98.38%** |
| | 0.2 | 56.82% | 50.85% | 66.96% | 66.72% | 52.65% | 52.99% | 67.00% | **96.94%** |
| DPMA | 0.1 | 56.94% | 50.61% | 70.63% | 68.14% | 52.78% | 52.90% | 71.12% | **97.63%** |
| | 0.15 | 55.87% | 50.60% | 64.54% | 63.64% | 52.70% | 52.23% | 64.47% | **96.74%** |
| | 0.2 | 54.37% | 50.51% | 58.90% | 58.36% | 52.76% | 51.37% | 60.03% | **94.59%** |

**Table 4: Extraction accuracy with different LDM versions.**

| Versions | | v1.2 | v1.4 | v1.5 | v2.1 | Deli | DS | RV |
|----------|------|------|------|------|------|------|------|------|
| Eular | Acc | 98.16% | 98.18% | 98.25% | 98.30% | 98.37% | 98.31% | 98.20% |
| | PSNR | 22.58 | 22.63 | 22.54 | 23.06 | 22.93 | 22.61 | 22.99 |
| LMS | Acc | 98.56% | 98.26% | 98.38% | 98.63% | 98.47% | 98.41% | 98.31% |
| | PSNR | 22.29 | 22.22 | 22.78 | 22.86 | 22.33 | 22.99 | 22.93 |
| DPMA | Acc | 96.44% | 96.54% | 96.74% | 97.04% | 96.56% | 96.52% | 96.51% |
| | PSNR | 21.41 | 21.50 | 21.40 | 21.74 | 21.41 | 21.90 | 21.56 |

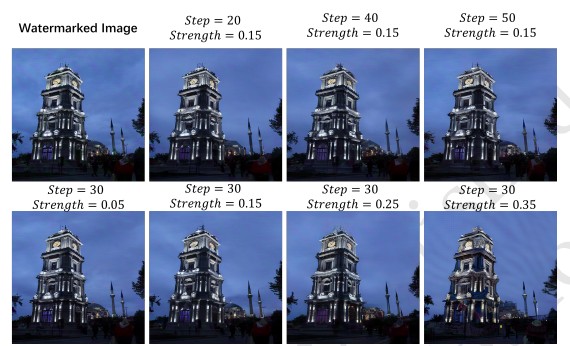

**Figure 6: The visual results of the watermarked image with sampling steps and denoising strengths.**

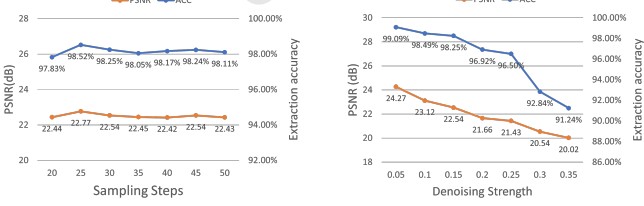

(a) Influence of sampling steps.     (b) Influence of denoising strength.

**Figure 7: The influence of sampling steps and denoising strengths in LDE.**

denoising strengths ranging from 0.05 to 0.35. The visual results of the erased images are presented in Fig. 6. Furthermore, we showcase the corresponding PSNR of the erased images and extraction accuracy with different erasure settings in Fig. 7.

From Fig. 6, it is apparent that the most influential parameter is the denoising strength. As the denoising strength increases, there is a significant alteration in the details of the images. On the other

hand, variations in the sampling steps do not significantly affect the erasure performance of the images. Since sampling steps only impact the sampling schedules and not the starting point, they do not produce substantial differences in the final image. These observations are also reflected in the results of PSNR values and extraction accuracy, as shown in Fig. 7. When fixing the denoising strength and increasing the sampling steps, the PSNR of the erased images remains consistent, as does the extraction accuracy. However, when fixing the sampling steps and increasing the denoising strength, the visual quality of the erased image deteriorates, evidenced by a decrease in PSNR. Similarly, the extraction accuracy also decreases with increasing denoising strength. It is worth noting that even when the denoising strength is set to 0.35, resulting in heavily altered erased images, the extraction accuracy remains higher than 91%. This indicates the strong robustness of the proposed framework against LDE distortions.

*5.4.3 Different Guidance Scales.* In latent diffusion model generation, to produce an image that aligns with conditioning $C$ to the desired extent, the model must exhibit a stronger bias toward generating outputs that are in harmony with $C$. This bias is typically accomplished by adding weights to the unconditional prediction $\epsilon_\theta (x_t, t, \varnothing)$. Thus, the final generation process is expressed as:

$$\Phi_\theta (x_t, t, C, G) = \epsilon_\theta (x_t, t, \varnothing) + G \times (\epsilon_\theta (x_t, t, C) - \epsilon_\theta (x_t, t, \varnothing))$$

where $G$ is the weight called the guidance scale. Different $G$ will result in different generation effects. In this section, we test the LDE robustness of the algorithm under different settings of $G = \{1, 3, 5, 7, 9, 11, 13\}$. The test model version is v1.5, with fixed sampling steps at 30, denoising strength at 0.15, and the sampling method "Eular". The results are presented in Fig. 8 and Table 5.

**Table 5: Extraction accuracy with different guidance scales.**

| $G$ | 1 | 3 | 5 | 7 | 9 | 11 | 13 |
|-----|------|------|------|------|------|------|------|
| Acc | 98.18% | 98.20% | 98.35% | 98.25% | 98.12% | 98.06% | 98.26% |
| PSNR | 22.61 | 22.66 | 22.47 | 22.54 | 22.59 | 22.56 | 22.58 |

It can be seen that the changing of the guidance scale will not greatly influence the visual quality of the erased images. Even with guidance $G = 11$, the images are still similar to those with guidance $G = 1$, where the PSNR value with different guidance scales is at the same level. This phenomenon occurs because the LDE process adds only a small amount of noise to the original latent

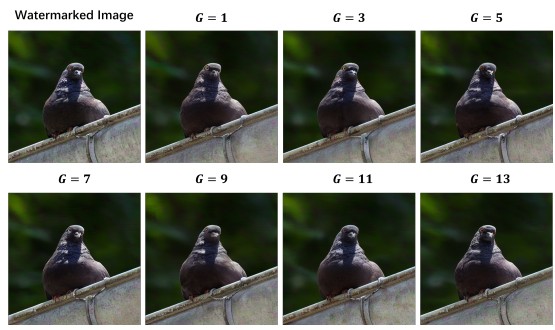

**Figure 8: The visual results of the erased image with different guidance scales.**

as the starting point, rather than generating from random noise. Additionally, the corresponding timestep weights are not very large. Consequently, the guidance scale does not have a substantial impact on the generated image. Furthermore, from Table 5, it is apparent that for all tested guidance scales, the proposed method achieves high extraction accuracy exceeding 98%. This demonstrates the robustness of the algorithm across different guidance scales.

## 5.5 Compatibility of Robustness

It is crucial to consider that in real-world scenarios, watermarked images may encounter various distortions beyond just LDE. Therefore, assessing the robustness compatibility of the proposed framework to these different distortions is essential. In this section, we conduct experiments to evaluate the framework's compatibility with various distortions. Following the common settings of state-of-the-art methods [6, 15], we train the entire framework with a combined noise layer containing Gaussian noise, JPEG compression, Dropout and Gaussian blur, named DERO++. Subsequently, we test the corresponding robustness with DERO and DERO++. The results of these experiments are presented in Table 6.

**Table 6: Extraction accuracy with different distortions.**

| Distortions | Gaussian Noise | | | Salt& Pepper Noise | | |
|---|---|---|---|---|---|---|
| | $\sigma^2 =0.01$ | 0.02 | 0.05 | $r =0.01$ | 0.02 | 0.03 |
| DERO | 98.41% | 97.31% | 94.57% | 98.06% | 96.78% | 95.22% |
| DERO++ | **99.58%** | **99.40%** | **98.89%** | **98.89%** | **97.80%** | **97.60%** |

| Distortion | Gaussian Blur | | | Median Blur | | |
|---|---|---|---|---|---|---|
| | $\sigma^2 =1$ | 2 | 3 | $w =3$ | 5 | 7 |
| DERO | 94.85% | 94.98% | 95.34% | 98.93% | 97.81% | 96.05% |
| DERO++ | **99.76%** | **99.81%** | **99.85%** | **99.67%** | **99.79%** | **99.80%** |

| Distortion | JPEG Compression | | | Dropout | | |
|---|---|---|---|---|---|---|
| | QF=50 | 60 | 70 | p=0.7 | 0.8 | 0.9 |
| DERO | 93.13% | 95.77% | 97.78% | 79.70% | 84.06% | 89.22% |
| DERO++ | **99.15%** | **99.23%** | **99.37%** | **91.19%** | **94.18%** | **96.50%** |

Table 6 highlights the strong robustness compatibility of DERO++, with extraction accuracy exceeding 91% for all tested distortions. When only with DERO, the robustness with certain distortions is not strong enough. Taking "Dropout" distortion as an example, the extraction accuracy of DERO against "Dropout-0.08" is only 84.06%. But with combined noise layer training, it achieves 94.18%. Remarkably, the extraction results under "Median Blur", "Salt & Pepper Noise" demonstrate the generalizability of the proposed

method. These positive results underscore the effectiveness of the designed DCNL in cooperation with traditional noise layers.

## 5.6 Ablation Study

*5.6.1 Importance of DCNL and LFA.* The crucial component for the LDE robustness of the proposed method is the design of DCNL and LFA. To demonstrate the importance of each, we conduct ablation experiments. We train the whole model with only DCNL and LFA individually. Additionally, we attempt a trivial approach to simulate LDE distortion by generating massive "original-distortion" image pairs and training a neural network to mimic the distortion. This neural network can also serve as the noise layer for training. Specifically, we generate 5000 training pairs using MS COCO training datasets [13] (Stable Diffusion v1.5, sampling steps 30, denoising strength 0.15, guidance scale 7, sampling method "Eular"). Then, we utilize the network architecture proposed in [34] to train the surrogate model for LDE distortions and train it with the well-trained distortion simulation model. Finally, we test the LDE robustness with these three models. The results are shown in Table 7.

**Table 7: Extraction accuracy with different noise layers.**

| Settings | Surrogate | *DCNL* | *LFA* | *DCNL + LFA* |
|---|---|---|---|---|
| Eular | 81.45% | 95.01% | 95.81% | **98.25%** |
| LMS | 81.41% | 95.47% | 94.76% | **98.38%** |
| DPMA | 78.61% | 90.93% | 93.03% | **96.74%** |

For a fair comparison, the PSNR of the watermarked images trained with different settings is set at the same level as 40.5 ± 0.2 dB. It can be seen that when training with the surrogate model, the extraction accuracy can only achieve 81%, which is 15% lower than the proposed methods. We believe the reason is that a simple network cannot well simulate the LDE distortion since the LDE process is complex. On the other hand, training only with DCNL and LFA ensures a certain level of robustness, with an extraction accuracy of up to 95%. Both DCNL and LFA simulate the LDE distortion to some extent, contributing to this level of robustness. However, training with both DCNL and LFA achieves the best extraction accuracy, surpassing the other settings by 2%. This indicates that the combination of DCNL and LFA synergistically enhances the model's ability to LDE robustness.

## 6 CONCLUSION

This paper points out a potential vulnerability in watermarking systems, revealing their susceptibility to high-quality erasure using the latent diffusion model. While existing schemes mainly address image processing distortions, they often fall short in adapting to latent diffusion erasure distortion. To fortify robustness, we first analyze the denoising process of LDM in the frequency domain. Leveraging the observation of the generation in frequency components, we successfully provide a destruction and compensation noise layer, which gives a rough simulation of LDE distortions. Combined with a VAE-based decoder and a latent feature augmentation operation, the whole system can be trained to guarantee LDE robustness. Experimental results affirm the method's efficacy, demonstrating superior robustness against latent diffusion erasure compared to state-of-the-art methods.

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
