# OpenReview forum: "DERO: Diffusion-Model-Erasure Robust Watermarking"
_acmmm.org/ACMMM/2024/Conference — MM2024 Poster_

### Official Review · Reviewer_Hvsu · 2024-05-20

**Rating:** 4
**Confidence:** 3

**Summary:**

This paper addresses the challenge of image watermark robustness in the face of advanced denoising attacks using latent diffusion models (LDM). Traditional watermarking methods are ineffective against such attacks, where attackers can effectively remove watermarks through forward diffusion followed by backward denoising. This paper proposes a diffusion-model-erasure robust watermarking framework called DERO. DERO simulates the distortions caused by LDM erosion using frequency domain analysis to design a destruction and compensation noise layer (DCNL). Additionally, a pre-trained variational autoencoder is cascaded before the decoder to extract the watermark in the latent domain, enhancing the system's robustness. DERO significantly improves robustness against LDM erosion, with experimental results showing an increase in robustness from 75% to 96% on the MS COCO dataset compared to state-of-the-art methods.

**Strengths:**

**Strengths:**

1. First of all, this paper is well-written and structured clearly, making it easy for anyone to read and understand.
2. The design of the DCNL part is very interesting. It cleverly uses high- and low-frequency information to simulate LDM attacks.
3. Overall, this paper provides a high-quality watermark embedding method that resists diffusion erasure without compromising image quality (even slightly improving it), offering a new and effective way for copyright protection.
4. The overall experimental design is very reasonable. The visual quality experiments demonstrate that the method does not degrade the quality of the embedded watermark. The robustness against LDE experiments proves the method's effectiveness in countering diffusion erasure. The ablation experiments show the importance of the proposed DCNL and LFA methods.
5. The supplementary materials provide more visual results, further proving the superiority of the method.
6. In this paper, the authors just use binary messages as the watermark. Is there any other option? What are the challenges or benefits?

**Limitations:**

**Weaknesses:**

1. My biggest concern comes from the comparison with state-of-the-art methods. The paper [ZoDiac: Robust Image Watermarking using Stable Diffusion](https://arxiv.org/abs/2401.04247v1) undertakes almost the same task as the authors and claims to be the first work to counteract diffusion watermark removal, showing seemingly better results. I suggest the authors compare their method with ZoDiac under fair settings and update the results accordingly.

2. Another concern is that the authors conducted experiments only on the [MS-COCO](https://arxiv.org/abs/1405.0312) dataset. I recommend that the authors also perform the same experiments on other datasets such as [DiffusionDB](https://arxiv.org/abs/2210.14896) and [WikiArt](https://arxiv.org/abs/1505.00855v1).

3. The authors mentioned the GPU used in their experiments but did not provide details on computational complexity, training time, and inference time.

4. The diffusion models used in this paper are mainly applied to generation tasks. Have the authors tried adversarial experiments against diffusion models specifically designed for watermark attacks, such as [Evading Watermark-based Detection of AI-Generated Content](https://arxiv.org/abs/2305.03807) and [Invisible Image Watermarks Are Provably Removable Using Generative AI](https://arxiv.org/abs/2306.01953)?

5. Just curious, could the authors explain why the watermark image quality generated by DERO is higher than that of other watermarking methods? Which part of the process enhances the quality of the embedded watermark?

ZoDiac can be considered concurrent work to DERO, so it is understandable that a comparison was not made. Given the high quality of the authors' work, I am giving a score of Borderline Accept. If the authors can supplement their work with more experiments, I promise to increase the score. However, if my concerns are not addressed, I will lower the score.

**Suitability:**

3

---

### Official Review · Reviewer_5aKG · 2024-05-25

**Rating:** 3
**Confidence:** 4

**Summary:**

This paper proposes a watermarking framework to improve the robustness of watermarking to diffusion models, called DERO. DERO incorporates a Destruction and Compensation Noise Layer (DCNL) to simulate the distortion effects of the Latent Diffusion Model Erasure (LDE) and a Latent Feature Augmentation (LFA) operation to enhance the robustness of the watermark extraction process.

**Strengths:**

This manuscript proposes an interesting watermarking scheme to resist the diffusion-model-erasure attack. The paper is well-structured, and the proposed scheme is clearly described. In the experiments, the proposed method demonstrates superior visual quality and robustness under LDE distortions.

**Limitations:**

The experimental evaluation could be further improved.
1.  The article considers diffusion and denoising processes in LDE attack. As far as I know, this is not a common attack. However, image editing based on diffusion models is more commonly used, such as image inpainting and outpainting[1].
2. Although the proposal is mainly used to improve the robustness of LDE, common attacks should also be considered, such as geometric attacks (resize, crop, rotation, flip) [2] [3].
3. The method only compares the robustness with other methods on the LDE attack. In Table 6, the proposed method is not compared with previous methods in terms of more common distortion attacks.

[1] Fernandez, Pierre, et al. "The stable signature: Rooting watermarks in latent diffusion models." Proceedings of the IEEE/CVF International Conference on Computer Vision. 2023.

[2] Ma, Rui, et al. "Towards blind watermarking: Combining invertible and non-invertible mechanisms." Proceedings of the 30th ACM International Conference on Multimedia. 2022.

[3] Guo, Hengchang, et al. "Practical Deep Dispersed Watermarking with Synchronization and Fusion." Proceedings of the 31st ACM International Conference on Multimedia. 2023.

**Suitability:**

2

---

### Official Review · Reviewer_ztvE · 2024-05-25

**Rating:** 5
**Confidence:** 4

**Summary:**

The paper proposes a robust watermarking method which grants resilience against diffusion model erasure.

**Strengths:**

The paper is well structured and presented.
It addresses the issue of robustness in deep learning-based watermarking.
Performances seems to improve the sota methods in terms both of visual quality and watermark extraction accuracy.

**Limitations:**

Some experimental details are missing and an analysis on the various issues of the procedure would be needed as evidenced by the major comments hereafter:

1) It is not so clear if the method can maintain the same performance when the watermarked image does not belong to the same dataset used for training (MS COCO in this case), could the authors discuss such an issue? This is crucial because it determines the generalization capability of the proposed procedure.

2) Figure 4, the row "residual": it is not understandable how residual image is computed. Is it computed as difference between the host image and the watermarked one or between the watermarked and erased one?

3) There is nothing concerning the impact of the watermark message: which is the length of the watermark? How many bits? What about information capacity?

4) The sota methods: have they been retrained or taken as they were? What about the set-up for the comparison?

**Suitability:**

3

---

### Meta-Review · Area_Chair_kYkQ · 2024-07-01

**Recommendation:** Accept (Poster)
**Confidence:** 5

**Metareview:**

This manuscript is well-written and provides convincing evaluations.
The unclear parts also have been clarified during the rebuttal process.
ALL reviewers consistently gave positive comments on this paper.
Thus, the AC recommends to accept this manuscript.